Protein function prediction with gene ontology: from traditional to deep learning models

Vu Thi Thuy Duong
Jung Jaehee jhjung@mju.ac.kr
Department of Information and Communication Engineering, Myongji University , Yongin-si , Gyeonggi-do , South Korea
Uversky Vladimir
Electronic publication date: 2021 Aug 24
Publication date: 2021
Volume: 9
Electronic Location ID: e12019
Received 2021 Apr 13; Accepted 2021 Jul 29
Copyright: ©2021 Vu and Jung
Copyright year: 2021
Copyright holder: Vu and Jung
License: This is an open access article distributed under the terms of the Creative Commons Attribution License, which permits unrestricted use, distribution, reproduction and adaptation in any medium and for any purpose provided that it is properly attributed. For attribution, the original author(s), title, publication source (PeerJ) and either DOI or URL of the article must be cited.
License URL: https://creativecommons.org/licenses/by/4.0/

Keywords: Gene Ontology, Protein function prediction, Machine learning, Deep learning, CAFA3, Annotation

Funding: The National Research Foundation of Korea (NRF) funded by the Ministry of Science, ICT and Future Planning NRF-2019R1A2C1084308 This research was supported by the National Research Foundation of Korea (NRF) funded by the Ministry of Science, ICT and Future Planning (NRF-2019R1A2C1084308). The funders had no role in study design, data collection and analysis, decision to publish, or preparation of the manuscript.

==============================
Protein function prediction is a crucial part of genome annotation. Prediction methods have recently witnessed rapid development, owing to the emergence of high-throughput sequencing technologies. Among the available databases for identifying protein function terms, Gene Ontology (GO) is an important resource that describes the functional properties of proteins. Researchers are employing various approaches to efficiently predict the GO terms. Meanwhile, deep learning, a fast-evolving discipline in data-driven approach, exhibits impressive potential with respect to assigning GO terms to amino acid sequences. Herein, we reviewed the currently available computational GO annotation methods for proteins, ranging from conventional to deep learning approach. Further, we selected some suitable predictors from among the reviewed tools and conducted a mini comparison of their performance using a worldwide challenge dataset. Finally, we discussed the remaining major challenges in the field, and emphasized the future directions for protein function prediction with GO.

Introduction

Proteins are organic macromolecules that are fundamental determinants of the structure and function of living organisms. They play a role in numerous processes, including biochemical reactions, transmission of signals, nutrient transport, immune system boosting, etc. Therefore, understanding protein properties is essential not only from the biological and evolutionary perspectives, but also from the viewpoint of leveraging their potential in biomedical and pharmaceutical applications, and other areas (Amiri-Dashatan et al., 2018).

Generally, protein function identification is accomplished through manual or computational annotation. The former approach is the gold standard for functional annotation, because it is implemented by expert annotators and yields high quality curated results. Nonetheless, this approach is expensive and laborious, and thus, it is difficult to scale. Furthermore, because of the development of high-throughput sequencing technologies, such as next-generation sequencing (NGS), the amount of sequences to be annotated has increased dramatically. Thus, computational annotation methods have been developed as a must to automatically process the high volume of newly generated sequences, and also to improve the accuracy of the annotated data (Sleator & Walsh, 2010).

Because of the variability in the vocabulary used to define protein function, which makes the annotation process confusing to both humans and machines, various databases have been proposed to provide a standardized scheme, such as the Enzyme Commission (EC) (Webb, 1992), Functional Catalogue (FunCat) (Ruepp et al., 2004), and Kyoto Encyclopedia of Genes and Genomes (KEGG) (Kanehisa et al., 2004). Currently, Gene Ontology (GO) (Ashburner et al., 2000; Consortium, 2015) is the most comprehensive resource, as it possesses all the desirable properties of a functional classification system (Pandey, Kumar & Steinbach, 2006). The GO consortium created a database for a controlled vocabulary describing the functional properties of genomic products (e.g., genes, proteins, and RNA). Each ontology (vocabulary) belongs to one of three categories: molecular function (MF), biological process (BP), and cellular component (CC). In terms of structure, the GO follows a hierarchical organization as a directed acyclic graph (DAG), in which each term is a node and each edge connected to two nodes represents a parent–child relationship. It can be used to infer many types of information, such as “is-a” or “part-of”. For instance, if term A is denoted as “is-a” of term B, it means that A is a sub-type of B. Further, “part-of” implies that the child node is necessarily part of the parent. This allows the flexible annotation of proteins with respect to the various levels of function—from general to specific terms—depending on the available evidence (Stein, 2001).

Automated function prediction (AFP) based on the GO system is a challenging problem in bioinformatics. Many studies have discussed protein functional annotation from different perspectives. Previous reviews focused on AFP (Rost et al., 2003) in terms of the data type used (Watson, Laskowski & Thornton, 2005; Pandey, Kumar & Steinbach, 2006; Sleator & Walsh, 2010; Shehu, Barbará & Molloy, 2016), drawbacks and corresponding solutions (Friedberg, 2006), protein interaction networks (Sharan, Ulitsky & Shamir, 2007), types of classified function (Rentzsch & Orengo, 2009), and GO assignment based on sequence information (Cozzetto & Jones, 2017). The two latest reviews are prepared by Bonetta & Valentino (2020) and Zhao et al. (2020). Bonetta & Valentino (2020) demonstrate protein function prediction in the machine learning workflow. Meanwhile, Zhao et al. (2020) discuss prediction of gene function prediction from the GO modeling perspective. All these researches provided independent perspectives to the issue, but we observed that there has not been any detailed review about deep learning, which is an emerging approach for protein function prediction with GO terms. Therefore, our paper suggests the possibility of predicting protein function using both conventional learning and deep learning, further indicating that better predictive performance can be expected by comparing several methods with each other.

Herein, we reviewed automated protein function prediction using GO terms (ranging from traditional solutions to the most recently developed deep learning-based tools). We presented not only an overview of the literature, but also, a performance comparison of the emerging solutions. Further, we highlighted the challenges and future prospects of the field. We hope that this review will provide bioinformaticians, computer scientists or any researchers who are interested in this topic, with the latest updates in terms of protein features used, models, and evaluation criteria, hoping to contribute to further improvements in the future.

Survey Methodology

Based on previous studies (Jung & Thon, 2006; Jung et al., 2010) and the aforementioned surveys, our review is divided in two main parts offering an overview of the field (from its inception to its present state). The first parts covers the conventional approach and includes solutions that do not use deep learning, while the second part describes methods that rely on deep learning to address the problem of protein functional annotation. We used Google Scholar https://scholar.google.com/ as the literature database to retrieve relevant publications, without applying any restrictions as to the publishing data, journal, or publisher.

Below, we review the conventional approach briefly, as some excellent literature reviews are already available on the subject (Shehu, Barbará & Molloy, 2016; Cozzetto & Jones, 2017). In that part, we summarize three main sub-categories of the traditional approach, and mention prominent or most recent corresponding studies. The second part of the review is our main focus. Search results using combinations of the following keywords: “deep learning”, “deep neural network”, “Gene Ontology”, “GO terms”, “protein function prediction”, “functional annotation”, and “gene function prediction”, are selected after screening the titles, abstracts and conclusions. We also referred to references cited in the downloaded papers, to capture significant studies in the field.

Conventional Approach for Predicting Protein GO Terms

In general, several techniques have been proposed to tackle the assignment of GO terms to proteins using different types of data and methods. We review the representative solutions below, focusing on three categories: similarity-based methods, probabilistic methods, and machine learning methods.

Similarity-based methods

Initially, functional annotations were assigned to uncharacterized proteins based on a simple principle: a protein sequence was searched in databases of experimentally curated proteins and if there were any similar proteins retrieved (“similar” in a specifically defined way), GO terms associated with the retrieved sequences were assigned to the query protein.

The very first solution of similarity-based methods was homology-based. These tools relied on local alignment search tools, such as the Basic Local Alignment Search Tool (BLAST) (Altschul et al., 1997). In these methods, the unknown sequence is searched in a database that curates well-annotated proteins. Then, a retrieved sequence with the highest alignment score—as per a predetermined threshold—is identified and its annotation is transferred to the query protein. OntoBlast (Zehetner, 2003), GOFigure (Khan et al., 2003), GOblet (Hennig, Groth & Lehrach, 2003), and GOtcha (Martin, Berriman & Barton, 2004) are typical annotation systems adopting sequence similarity determined by BLAST search. Further details regarding these tools are presented in Table 1.

Table 1 GO annotation predictors using the traditional approach.

Methods	Program name	Description	Reference	
Similarity based	OntoBlast	Web server, which is a part of the “Ontologies TO GenomeMatrix” tool (Hewelt et al., 2002). It predicts protein function by a weighted list of GO terms associated with BLAST hit sequences in nine genome databases.	Zehetner (2003)	
GOFigure	Produces output as a clickable graph in four steps, including homologous sequence search, minimum cover graph construction, and assigning ontologies after scoring them.	Khan et al. (2003)	
	GOblet	Software package allowing the user to define the E-value sensitivity and databases used for the BLAST search. All ontologies identified after matching the proteins are constructed in a summary DAG structure, where the number of sequences sharing a common GO term is cumulative to present the significance of the term.	Hennig, Groth & Lehrach (2003)	
	Gotcha	Transfers GO associations retrieved from BLAST to gene products through a novel ranking scheme. The E-values of the derived terms are propagated to their parent terms on DAG, providing normalized confidence scores for an individual GO ontology.	Martin, Berriman & Barton (2004)	
	PFP	Functional annotation system based on PSI-BLAST hits, exploiting matched sequences up to an unconventionally high threshold, and mining highly relevant GO terms from UniProt to score functions for unknown proteins.	Hawkins et al. (2009)	
	INGA	Functional annotation server derives GO predictions from combining many data sources.	Piovesan et al. (2015)	
	GoFDR	Multiple alignment-based method utilizing FDRs and PSSM to rank GO terms for amino acid sequences.	Gong, Ning & Tian (2016)	
Probabilistic	–	Probabilistic framework based on PPI, predicting GO assignments for 10% unannotated proteins in yeast.	Letovsky & Kasif (2003)	
	–	Leverages heterogeneous data to increase the probabilistic based model for the functional prediction of Saccharomyces cerevisiae.	Nariai, Kolaczyk & Kasif (2007)	
	BMRF	Variant of the MRF-based approach on PPI that infers novel predictions for unannotated proteins.	Kourmpetis et al. (2010)	
Machine learning	GOPET	GO term prediction and evaluation tool, supplying molecular function and biological process GO terms for any organism.	Vinayagam et al. (2006)	
	PoGO	Ensemble model with InterPro sequence similarity, biochemical property, and protein tertiary structure to predict the GO of fungal proteins.	Jung et al. (2010)	
	FFPred3	SVM-based tool, providing the homology-independent assignment of GO terms for eukaryotic proteins.		
	PANNZER2	Weighted k-nearest neighbor classifier, providing functional annotation for uncharacterized proteins with GO terms and free text descriptions.	Törönen, Medlar & Holm (2018)	
	DeepText2GO	Combination of a text-based method and a sequence-based method to improve large-scale protein function prediction.	You, Huang & Zhu (2018a)	
	NetGo	Effective LTR-based web server, combining both sequence and massive network information of proteins to annotate gene products.	You et al. (2019)	

PFP (Hawkins et al., 2009) is another method that utilizes functional information associated with remote homologs by employing Position-Specific Iterated BLAST (PSI-BLAST) (Altschul et al., 1997). The tool is an improvement in terms of coverage and accuracy, as determined by analyzing a benchmark dataset, compared with GOtcha, top PSI-BLAST, and InterProScan (Zdobnov & Apweiler, 2001). Another variant solution based on sequence alignment is INGA (Piovesan et al., 2015), in which protein–protein interaction (PPI) network data is combined with domain assignment and sequence similarity from BLAST, to attain a consensus prediction of GO functions using enrichment analysis. Further, GoFDR (Gong, Ning & Tian, 2016) takes relevant GO terms from multiple sequence alignment (MSA) querying via BLAST or PSI-BLAST search. The probability of assigning a term to the query sequence is determined by the functionally discriminating residues (FDRs), a position-specific scoring matrix (PSSM) for the FDRs, and a score-to-probability table prepared using training sequences.

Although methods based on a local alignment search are straightforward and perform well to some extent, they also have some drawbacks, including database annotation errors or excessive function transfer, threshold relativity, and low sensitivity or specificity (Sasson, Kaplan & Linial, 2006). In addition to the alignment-based methods, several other predictors are available that transfer function annotations based on the similarity at the level of protein structure, protein family, or phylogenomics (Shehu, Barbará & Molloy, 2016).

Probabilistic methods

A number of probabilistic models have been devised for protein function deduction (Deng, Chen & Sun, 2004a; Deng et al., 2004b; Letovsky & Kasif, 2003; Nariai, Kolaczyk & Kasif, 2007). Letovsky & Kasif (2003) employed a functional linkage graph (Marcotte et al., 1999) constructed based on a PPI network of the yeast Saccharomyces cerevisiae. The working assumption was that the probability of sharing functions between (nodes) proteins in close proximity on the graph is higher than that for nodes that are not in close proximity. In this method, the probabilities of GO terms assigned to a protein sequence are derived from a binomial model incorporating the Markov Random Field (MRF) algorithm. Later, Nariai, Kolaczyk & Kasif (2007) integrated multiple sources of evidence (PPI, gene expression data, protein motif information, mutant phenotype data, and protein localization data) using a Bayesian framework, to improve the prediction performance, compared with their model that only uses PPI.

Working on the same model organism, S. cerevisiae, as used in the methods described above, Kourmpetis et al. (2010) suggested applying a Bayesian approach to the MRF model. This suggestion was considered an improvement for the estimation of model parameters and the provision of predictions from network data. Further, Pinoli, Chicco & Masseroli (2015) compared different weighting schemes as combined with three algorithms, namely, truncated singular value decomposition (tSVD), semantically improved tSVD (SIM), and probabilistic latent semantic analysis with normalization (pLSAnorm). The first two methods are based on linear algebra, while the latter is a modified probabilistic model based on Bayesian inference. These methods were successfully used to generate novel GO annotations for three model organisms, i.e., Bos taurus, Danio rerio, and Drosophila melanogaster.

Machine learning methods

Machine learning-based tools have been developed to identify the hidden relationships between various protein features (sequence, structure, or other related evolutionary evidence) and functional labels, based on a training set (a group of fully characterized macromolecules), and use that information to generate annotations for novel proteins. Various methods that rely on machine learning have been suggested, incorporating different lines of evidence as features to train the classifiers and predict GO terms. Branch (a) of Fig. 1 formulates the common workflow of those solutions, the details of which can vary between methods as described below.

Figure 1 A common workflow of (A) machine learning-based and (B) deep learning-based solutions for predicting the GO terms of proteins.

The components presented in the dashed box are optional, depending on each method. N is the number of GO terms that are selected for the training dataset to build the model.

Utilizing multiple support vector machine (SVM) classifiers is a predominant selection in several studies. For instance, GOPET (Vinayagam et al., 2006) feeds several features related to GO terms (sequence similarity measures based on BLAST search, frequency, quality of homolog annotation, and annotation level within the GO hierarchy) to 99 SVM classifiers. A specific term will be predicted to be a “correct” or “incorrect” label for an unknown sequence, with voting confidence scores calculated. As another example, FFPred (Lobley et al., 2007; Lobley et al., 2008) was initially established for the unannotated human proteome, but demonstrated a generalization to other proteomes. The latest version of FFPred is FFPred3 (Cozzetto et al., 2016), which is still based on SVM, is expanded to investigate the correlations between feature characteristics retrieved from sequences and structures within the three sub-ontologies (MF, BP, and CC). Another tool, Prediction of Gene Ontology terms (PoGO) (Jung et al., 2010), has been developed from Automatic Annotation of Protein Functional Class (AAPFC) (Jung & Thon, 2006). Instead of only using InterPro terms as features, PoGO integrates three more sources (sequence similarity, biochemical properties, and protein tertiary structure). Subsequently, SVM and a linear classifier (Freund & Schapire, 1997) are used as base-level classifiers before the meta-learning step.

Using the k-nearest neighbor (k-NN) algorithm, PANNZER2 (Törönen, Medlar & Holm, 2018) provides a fast functional annotation systems based on sequence homology and other annotation predictors. Meanwhile, MS-k NN (Lan et al., 2013) aggregates heterogeneous data to propose a competitive model for protein function prediction. DeepText2GO (You, Huang & Zhu, 2018a) is a consensus approach, integrating deep semantic text representation from MEDLINE citations (ncb, 2018) as text information, and sequence information obtained via BLAST and InterProScan. These features are fed to the k-NN and logistic regression models to produce the functions of no-knowledge proteins on a large scale. NetGO (You et al., 2019) is an extension of GoLabeler (You et al., 2018b), which employs the learning-to-rank(LTR) model to integrate sequence-based evidence. NetGO improves the performance of a large scale AFP by accessing the enormous protein–protein network of over 2000 species in the STRING database (Szklarczyk et al., 2016).

Machine learning approaches are highly advantageous and considered the future direction for AFP. Some novel proteins either lack identifiable homologous sequences or their detectable homologues have not been assigned any GO labels. Consequently, assigning protein function from scratch using machine learning, i.e., directly inferring the annotation from the amino acid sequence, without access to any additional references or databases, is an ongoing task. The sophisticated algorithms are being developed, as presented in the next section.

GO Term Annotation of Proteins Using Deep Learning Approach

The excellent potential of deep learning—the latest discipline of machine learning—has been demonstrated in several application fields, including bioinformatics (Min, Lee & Yoon, 2017; Li et al., 2019). The main characteristic that distinguishes this approach from other methods is the learning process. Deep learning models are inspired by neural networks in the human brain. They automatically extract high-level features from raw data and provide predictions in an end-to-end manner. In contrast to that in conventional machine learning models, accurate classification is achieved via handcrafted features.

Currently, the amount of generated genomic data, and the numbers of sophisticated algorithms and computational resources are rapidly growing. These resources support deep learning to tackle the functional annotation problem. Below, we review proposed methods based on two criteria, i.e., the model and the input data used. A summary table of the reviewed methods is presented in Table 2, and a common workflow for deep learning approach is illustrated as Figure 1B.

Table 2 Deep learning-based methods for assigning GO terms to proteins.

Feature used	Program name	Description	DL model used	
Sequence based	Chicco, Sadowski & Baldi (2014)	Deep AE-based method for GO term prediction.	AE	
	Zou, Wang & Yu (2017)	Infers missing GO annotations for proteins of Homo sapiens, S. cerevisiae, Mus musculus, and Drosophila using deep restricted Boltzmann machines.	DRBM	
	ProLanGO (Cao et al. (2017)	LSTM-based model using the idea of NMT for the functional annotation of amino acid sequences.	LSTM	
	SECLEF (Szalkai & Grolmusz, 2018b)	Generates the GO function and UniProt protein family from sequence data. The core models were developed by Szalkai & Grolmusz (2018a). The web-based interface is available at https://pitgroup.org/seclaf/.	CNN	
	DEEPred (Rifaioglu et al., 2019)	GO-based protein function prediction with a set of connected MTDNN. The source code and data are available at https://github.com/cansyl/DEEPred.	DNN	
	DeepSeq (Nauman et al., 2019)	Predicts H. sapiens protein function from sequence data only, using a deep learning architecture based on CNN. Source code and data available at https://github.com/recluze/deepseq.	CNN	
	DeepGOPlus (Kulmanov & Hoehndorf, 2020)	Advanced method of DeepGO, relying on a structure of several CNN layers. Available at http://deepgoplus.bio2vec.net/.	CNN	
Integrated data based/Structure based	Tavanaei et al. (2016)	Deep CNN model classifying functionality, based on the tertiary structure of human proteins.	CNN	
	DeepGO (Kulmanov, Khan & Hoehndorf, 2018)	Deep learning framework, employing one convolutional layer. Protein GO annotations are inferred based on a hierarchically structured classifier. Online tool available at https://deepgo.cbrc.kaust.edu.sa/deepgo/.	CNN	
	deepNF (Gligorijević, Barot & Bonneau, 2018)	Deep network fusion, captures high-level features of several network data for AFP. Source code and data available at https://github.com/VGligorijevic/deepNF.	AE	
	(Fa et al., 2018)	An implementation of two stacked multi-layer structures to predict GO terms from the sequence and structure properties. Source code and data available at http://bioinf.cs.ucl.ac.uk/downloads/mtdnn.	DNN	
	DeepFunc (Zhang et al., 2019)	Deduces Go annotations by concatenating InterPro-based and protein interaction network features.	FCDN	
	DeepGOA (Zhang et al., 2020)	Integrates comprehensive sequence and PPI information for AFP.	Bi-LSTM, CNN	
	SDN2GO (Cai, Wang & Deng, 2020)	Integrates architectures with three sub-models and a weighting classifier to achieve GO term predictions. Source code and data available at https://github.com/Charrick/SDN2GO.	CNN	
	DeepAdd (Du et al., 2020)	CNN-based framework, predicting protein function from sequence and additional information (PPI or SSP).	CNN	
	FFPred-GAN (Wan & Jones, 2020)	Novel FFPred that uses training sample augmented by a deep learning architecture.	GAN	
	GONET (Li et al., 2020)	Deep model for protein function prediction using representation learning to embed protein sequences and networks.	CNN, RNN	

Model-based approach

Supervised learning

Supervised learning is an essential approach for in silico protein function prediction. Supervised models learn to annotate function guided by a training dataset, which is a group of characterized proteins with reliable, experimentally confirmed annotations. After the learning stage, a model that captures the relationship between feature and function is produced, and used to predict the GO terms for novel amino acid sequences.

Deep neural network (DNN) is a feed-forward architecture. It generally comprises an input layer, multiple hidden layers, and an output layer. Input data are processed in a unidirectional manner along the layers, from the first to the final stage. With respect to GO-based functional annotation, DNNs have been employed as multi-task DNNs (MTDNNs) (Fa et al., 2018) or a series of MTDNNs (Rifaioglu et al., 2019).

Convolutional neural network (CNN) was initially developed to handle two-dimensional images for handwritten digit recognition (LeCun et al., 1990). However, it has since become an effective architecture not only for multi-dimensional data, but also for one-dimensional input, such as sentences or genomic sequences. A CNN model starts with convolutional layers, whose units are feature maps. Each unit is obtained by computing convolution operations between local patches of a unit in the preceding layer and a filter (set of weights) (LeCun, Bengio & Hinton, 2015). Non-linearity pooling layers are stacked to convolutional layers, enabling the integration of different-scale features in this architecture. CNNs can be applied in AFP either alone (Kulmanov, Khan & Hoehndorf, 2018; Cai, Wang & Deng, 2020; Du et al., 2020) or in combination with other architectures (Zhang et al., 2020; Spalević et al., 2020).

Recurrent neural network (RNN) is a deep learning architecture developed especially for sequential data. RNNs have a DNN backbone, and the units of the hidden layer are connected. Therefore, a hidden unit receives information from the input layer at the current time step, and also from the hidden units at the preceding stage (LeCun, Bengio & Hinton, 2015). Long short-term memory (LSTM) (Hochreiter & Schmidhuber, 1997) is a variant of the RNN model, developed to capture long-term dependencies between input sequences. Bi-directional LSTM (Bi-LSTM) is an LSTM model that processes data in two directions, forwards and backwards. These neural networks are employed to process protein sequences as a natural language (Cao et al., 2017) or are combined with the CNN model (Zhang et al., 2020) to provide GO annotations.

Another type of neural network, fully connected deep network (FCDN), is a series of fully connected layers. This model was utilized for converting input vectors from InterPro and predicting GO terms (Zhang et al., 2019).

Unsupervised learning

Unlike supervised learning, unsupervised learning independently mines hidden patterns in input distribution. It is crucial for exploring unlabeled data and provides important information for supervised architectures. Typically, unsupervised learning models are employed for clustering, reducing dimensions, and transforming data.

Autoencoder (AE) (Baldi, 2012) is an unsupervised learning model developed for assigning GO terms to amino acid sequences. AE is a neural network for data transformation. After encoding and decoding, the target output size is the same as that of the input. Using a trained AE model, GO predictions for proteins can be deduced directly from the generated matrix (Chicco, Sadowski & Baldi, 2014). Alternatively, low-dimensional feature representations in the hidden layer are extracted and fed into an SVM classifier for final classification (Gligorijević, Barot & Bonneau, 2018) or a CNN model (Peng et al., 2020) for final classification.

Restricted Boltzmann machine (RBM) (Salakhutdinov & Hinton, 2009) has one hidden layer for representing latent features and an input layer encoding the observed data. For GO annotation, units of the input layer are the available GO terms (Zou, Wang & Yu, 2017). Zou, Wang & Yu (2017) introduced a deep RBM (DRBM), where multi-layer RBMs are trained and unfolded, resulting in protein function prediction.

Finally, generative adversarial network (GAN) (Goodfellow et al., 2014) comprises two networks, i.e., a generative model that produces synthetic realistic data and a discriminative model that evaluates whether the data are real or not. In AFP, the GAN model improves the prediction by creating synthetic features for supervised classifiers (Wan & Jones, 2020), or by exploiting ontology correlations (Seyyedsalehi et al., 2021).

Data-based approach

In terms of the input feature, deep learning-based methods primarily use two approaches to assign GO terms to unknown gene products. One is a sequence-only approach, which is useful for predicting the functions of novel proteins in the absence of homologous information or other references. The second one is structure-based or otherwise exploits big data from several available resources.

Sequence-based models

One of the earliest deep learning-based methods for GO annotation predictions is the one proposed by Chicco, Sadowski & Baldi (2014). The authors compared two annotation solutions, i.e., tSVD, presented in Pinoli, Chicco & Masseroli (2013), and the AE neural network. The output form of the two methods was similar; however, the latter performed better on six different datasets. Later, instead of using the AE architecture, (Zou, Wang & Yu, 2017) proposed DRBM for annotating the gene products of four model species, Homo sapiens, S.cerevisiae, Musmusculus, and Drosophila.

ProLanGo (Cao et al., 2017) is the first tool that applies Neural Machine Translation (NMT) developed by Google in AFP. The authors converted amino acid sequences and GO terms into “ProLan” and “GOLan” languages, respectively. The GO annotation of a new protein was then generated after being translated using a model with three RNN layers. In another study, artificial neural networks proposed by Szalkai & Grolmusz (2018a) utilized six CNN layers of increasing depth. Based on that work, SECLEF (Szalkai & Grolmusz, 2018b) has been designed to train and test the GO function and UniProt protein family using biological sequences as input. The user can implement their self-configured models in a downloadable program or run two pre-trained models (Szalkai & Grolmusz, 2018a) on a web server. DeepSeq (Nauman et al., 2019) is another method for assigning GO terms to amino acid sequences, based on convolutional layers. However, the authors only predicted the five most frequent MF ontologies for H. sapiens proteins.

Using the DNN architecture, DEEPred (Rifaioglu et al., 2019) implements a stack of multi-task feed-forward DNNs. Each individual network is built for a specific GO term level on DAG, which allows the hierarchical post-processing of predictions. Protein sequences are experimentally represented by three types of descriptors(subsequence profile map, pseudo-amino acid composition, and conjoint triad feature), with subsequence profile map performing best in the analysis.

DeepGOPlus (Kulmanov & Hoehndorf, 2020) has been developed to overcome the existing limitations of DeepGO (Kulmanov, Khan & Hoehndorf, 2018), such as sequence length, unavailable PPI features, and the number of GO labels. Functional annotation is predicted by a multi-layer CNN structure in combination with sequence similarity. Lately, TALE (Cao & Shen, 2021) has been developed to generate GO predictions by integrating sequence patterns based on transformer encoder and the joint similarity of sequence-term. Similar to DeepGOPlus, TALE is also combined with sequence similarity as TALE+ model to enhance its performance. Meanwhile, PFP-WGAN (Seyyedsalehi et al., 2021) is one of two latest ideas that use GAN to infer the functionalities of proteins. While the generator network processes raw sequences, the discriminator takes two inputs; one consists of annotated proteins from the SwissProt database and the other of raw sequences with respective annotations synthesized from the generator.

Integrated data-based/Structure based-models

The functional assignment of amino acid sequences from 3D structures was proposed by (Tavanaei et al., 2016). The main analysis involved a CNN model that was trained and tested on five datasets of human proteins, each of which had been assigned two GO terms. However, the prediction has not been propagated on the DAG tree for inherited GO terms.

DeepGO (Kulmanov, Khan & Hoehndorf, 2018) is a functional annotation server based on CNN architecture that had been devised to learn features from amino acid sequences and the PPI network. GO labels are assigned to input proteins through a hierarchical classification structured as a DAG tree. DeepAdd (Du et al., 2020) was inspired by DeepGO server and provides a solution for AFP, utilizing a CNN framework to learn vector representations from sequences and additional information. However, sequences are processed into k-mer embedding by Word2Vec, instead of the tri-gram embedding suggested in DeepGO. Further, in DeepAdd, the protein sequence profile (SSP) is added if the PPI network information is not available. Based on a similar concept of combining primary protein structure and PPI, GONET (Li et al., 2020), a novel model, was built by employing CNN, RNN, and Attention layer for human and mouse sequences.

Working on a dataset identical to the one collated by FFPred3, Fa et al. (2018) tested an MTDNN that treated protein function prediction as a multi-label classification problem. Here, the solution consisted of layers shared by all tasks (GO labels), which are stacked in parallel with task-specific layers. DeepFunc (Zhang et al., 2019) is a novel predictor that surpasses DeepGO, FFPred3, and BLAST. First, protein domain, family, and motif information is queried from InterPro and encoded before passing through fully connected layers. Next, topological features of PPI are obtained by the Deepwalk algorithm. Finally, the method concatenates two types of features (sequence- and network-based input) to fit an FCDN. The architecture of DeepGOA (Zhang et al., 2020) is more sophisticated than that of DeepFunc. In addition to information generated by InterPro and PPI, global and local semantic features of amino acid sequences are extracted by Bi-LSTM and a convolutional layer, respectively. Using the same types of features, SDN2GO (Cai, Wang & Deng, 2020) utilizes three sub-models for each information source, with all outputs integrated in the final weighted model.

In another study, deepNF (Gligorijević, Barot & Bonneau, 2018) was constructed by multimodal deep AE to capture hidden information in proteins from different types of interaction networks. In that suggestion, feature representations are extracted to train the final SVM classifier to produce GO terms for human proteins. DeepMNE-CNN (Peng et al., 2020) has a superior performance than deepNF in the human data by utilizing CNN layers instead of SVM for the classification model. On the other hand, FFPred-GAN (Wan & Jones, 2020) utilizes GAN for feature enrichment to feed conventional machine learning models, especially SVM. Real features are biophysical information extracted from raw amino acid sequences by FFPred, and the generator takes latent variables to augment the synthetic samples.

Comparison

In addition to proposing a new approach, accuracy assessment is important for demonstrating the improvements of a novel methodology. This necessitates the establishment of a common framework for the evaluation and comparison of the proposed solutions. Accordingly, the critical assessment of functional annotation (CAFA) (Radivojac et al., 2013; Jiang et al., 2016; Zhou et al., 2019) is a community-based experiment that provides a large-scale evaluation of computational protein function prediction methods in a time-delayed manner. Four main competitions (CAFA1–CAFA4) have been held every 3 years since 2010. CAFA-π is an extension of CAFA3, and allows the competitors to improve their computational solutions focusing on the prediction of genes associated with specific GO terms. Generally, each challenge comprises three phases, i.e., predictions for a set of target proteins, experimental verification for a subset of target proteins to obtain a benchmark dataset, and performance analysis of all the competitors based on predefined metrics and the benchmark data. Official reports have just been released for the first three challenges and CAFA-π (Radivojac et al., 2013; Jiang et al., 2016; Zhou et al., 2019).

In the remainder of this section, we present a practical comparison of emerging analysis methods and discuss the retrieved results.

Set up

Data

We used the finalized benchmark set provided in the CAFA3 report (Zhou et al., 2019). The statistics are presented in Table 3. There are two sequence types in the benchmark dataset, NK (no-knowledge) and LK (limited-knowledge), which have been coined in CAFA2 (Jiang et al., 2016). Based on the timeline of the challenge, t0 is the deadline for the prediction submission of the target sequences. Then, experimental annotations for a subset of target sequences are accumulated until t1, to complete a benchmark dataset for the performance evaluation. The NK proteins are those that had not had any experimental annotations in three GO domains at t0, but had at least one GO term with experimental evidence code between t0 and t1. The LK sequences are those that had been experimentally annotated in one or two GO domains at t0, and gained at least one experimental annotation in the other GO domains between t0 and t1.

Table 3 Statistics of final benchmark CAFA3.

Annotation type	Number of sequences	Number of terms (GO leaf only)	
		MF	CC	BP	
NK (no-knowledge)	1177	377	236	1087	
LK (limited-knowledge)	2151	368	203	878	
All (NK + LK)	3328	631	327	1673	

Methods

The servers of several tools have been updated recently, but we need to control their training data or databases used, in which do not include sequences in the CAFA3 benchmark. Therefore, we chose INGA, which was one of the top models in CAFA3, as a representative conventional tool. DeepGOPlus is a recently developed representative annotation system that is based on deep learning. In addition to these two tools, other studies that performed their analysis on the CAFA3 dataset are also summarized in this part.

For DeepGOPlus, we re-trained the model with the CAFA3 training sequences and generated predictions for the CAFA3 benchmark set based on their source code provided. For INGA, we obtained predictions for the CAFA3 benchmark based on the predictions of CAFA3 targets, which were provided on their website.

Model performances were evaluated based on protein-centric metrics (Fmax, precision, and recall) provided by the CAFA3 assessment tool. The area under the precision–recall curve (AUPR), which is a measurement of highly imbalanced data, was also considered. Results regarding the performance of other methods, including Naive, BLAST, FFPred3 and DEEPred, were derived from DEEPred paper (Rifaioglu et al., 2019).

In terms of evaluation, two modes were designed. In the partial mode, the scores of each method were computed on the sequences that have at least one GO term predicted at any threshold. Conversely, scores in the full mode were computed on all protein sequences in the benchmark set.

Results

The results of the analysis are shown in terms of GO domains and types of sequences (NK vs. All) in full mode (Fig. 2) and partial mode (Fig. 3). The scores of Naive, BLAST, FFPred3 and DEEPred are only available for the full mode. Models using the traditional approach and deep learning approach are depicted in blue and orange tones, respectively.

Figure 2 Performance of the selected methods with the CAFA3 benchmark dataset in full mode: (A) NK sequences and (B) All sequences (NK + LK).

Scores in the full mode were computed on all protein sequences in the benchmark set. Scores of Naive, BLAST, FFPred3, DEEPred have been reported directly from Rifaioglu et al. (2019).

Figure 3 Performance of the selected methods with the CAFA3 benchmark dataset in partial mode: (A) NK sequences and (B) all sequences (NK + LK).

In the partial mode, scores of each method were computed on the sequences that have at least one GO term predicted at any threshold.

As is shown in Fig. 2, deep learning-based models (DEEPred and DeepGOPlus) showed a competitive performance compared to conventional models in terms of MF and BP prediction; in particular, DEEPred acquired the highest precision in the three GO categories. Such high precision obtained by DEEPred, which was explained in their paper (Rifaioglu et al., 2019), probably resulted from the imbalanced dataset and post-processing step. For example, the negative protein set for each GO term is four times as large as the positive one, this leads to low false positive and high false negative predictions. Additionally, DEEPred filtered some false positives at the post-processing step, thus increasing their precision. For the CC class, two baselines (Naive and BLAST) were at the top for Fmax and recall, followed by FFPred3. The overall outcome is presented similarly in Fig. 3, in which DeepGOPlus performed better than INGA in most of categories.

The aforementioned results and the CAFA3 report indicate that conventional models, such as INGA and GOLabeler (referred to as “ZhuLab” in CAFA3) perform as well as other high-performing models. ZhuLab was the best performer in terms of Fmax and NK-full evaluation, with the scores of 0.40, 0.61, and 0.62 for BP, CC, and MF, respectively. However, these conventional models integrate several types of information, which is not always available for all sequences. Meanwhile, the deep learning approach had demonstrable advantages in this area and yielded highly competitive predictions. The research direction is currently shifting toward this new approach, with even more efficient solutions being developed.

Conclusion

The computational GO annotation of proteins has been an actively pursued and challenging task in bioinformatics since around the 2000s; this is a response to the need to bridge the gap between the known and unknown, newly discovered amino acid sequences. On the one hand, understanding protein function is essential for deciphering biological evolution and for countless applications, such as drug design and disease treatment. The GO database has facilitated a comprehensive vocabulary for functional annotation because it presents structured GO functions in three domains (MF, BP, and CC), thereby effectively supporting in silico protein function assignment. Herein, we presented the current state of the field and compared GO-based AFP solutions, classed as conventional and deep learning approaches. As it is difficult to introduce all the available methodologies and comprehensively compare them, we tested the prominent predictors in this study. This was complemented by the analysis of related publications, with the assessment outcomes computed by CAFA, a worldwide venue for comparing computational protein function predictors.

We observed that one of the primary difficulties in the field is the utilization of input features to achieve effective performance. While heterogeneous input data are useful for GO prediction, only sequence information is available for the majority of unannotated proteins. With respect to the specific approaches, machine learning models may be limited by the heterogeneity of genomic data when mining different sources of information, while parameter and hyper-parameter tuning is the challenging step of the deep learning approach. In terms of output, the GO database has been updating because GO annotations are still imbalanced and not complete for all species. Its sophisticated structure is ideal for describing the functional roles of proteins, but also makes the prediction task a complex multi-label problem.

Nonetheless, despite the existing challenges, the quest for successful annotation will continue in many directions, driven by the community effort. Together with the constant expansion of -omics databases, data-driven methods have shown excellent applications in various areas, with a promising trend in functional annotation. Depending on the available sources, they can be based on integrated data or sequence information only, considering a hybrid approach. Some approaches have been suggested and developed to cope with imbalanced data; these include working on the subgroups in which the classes are more balanced, data augmentation using GAN, and considering evaluation metrics specified for imbalanced data, such as AUPR. The multi-label problem can be addressed via the advancement of computational resources and well-defined solutions, for example, stacking many individual solutions together. Finally, successful solutions for GO term prediction can be expanded to include other functional resources (EC, pathways, etc.), to comprehend the biological role and life science potential of proteins.

Additional Information and Declarations

Competing Interests

Author Contributions

Data Availability

The authors declare there are no competing interests.

Thi Thuy Duong Vu conceived and designed the experiments, performed the experiments, analyzed the data, prepared figures and/or tables, authored or reviewed drafts of the paper, and approved the final draft.

Jaehee Jung conceived and designed the experiments, analyzed the data, authored or reviewed drafts of the paper, and approved the final draft.

The following information was supplied regarding data availability:

Supplemental documents are available at GitHub: https://github.com/duongvtt96/Comparison-GO-annotation-systems

Dataset used in this comparison (final benchmark CAFA3) is available at Figshare:

Zhou, Naihui (2019): Supplementary_data. figshare. Dataset. https://doi.org/10.6084/m9.figshare.8135393.v3.

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
