# Peer review of "Protein function prediction with gene ontology: from traditional to deep learning models"

_PeerJ, doi:10.7717/peerj.12019_

## Round 0.1 · original submission · Major Revisions

Please address the critiques of both reviewers and revise the manuscript accordingly.

Reviewer 1 ·

Basic reporting

No comment

Experimental design

1 For the survey method part, I don’t think it is a fair comparison of the mentioned tools, since the benchmark set CAFA3 was published in 2019, the DeepGOplus was trained using more recent data released in 2020, and other tools, such as PFP and PANNZER2 were trained using different datasets (training data was not mentioned in the manuscript). The results of DEEPred were extracted from the DEEPred paper, the training data was still not clear. Because deep-learning models are easy to get the over-fitting problem, the sequence similarity between the training data and the benchmark set will strongly affect the performance. It is not clear whether the sequences in CAFA3 have been used in training those deep-learning methods, the performance may be overestimated for DeepGoplus and DEEPred. Since DeepGoplus and DEEPred both provide source code, they should be re-trained on the same dataset. Because of such a problem, the conclusions are not well supported. I suggest the authors do more strict experiments or discuss more on the performance evaluation.
2 For the machine-learning-based method, there lacks a clear problem formulation. For example, is it a multi-class classification problem for all the GO terms? Or is it a binary-class classification problem for each GO term? This is also related to the performance evaluation. The metrics used to evaluate the prediction performance, such as precision and recall, are designed for binary-class predictions, how to assign a single score per method?

Validity of the findings

1 Why only sequence-based methods were selected and compared in this paper? The readers expect to see the comparisons of deep-learning methods as much as possible.
2 As pointed out by the authors, one challenge of protein function prediction is the imbalanced GO classes and the multi-label problem, but what is the current state of the field regarding to these challenges? Do the authors have any insights in coping with these challenges?

Additional comments

In this review paper, the authors briefly reviewed the conventional approaches and focused on the review of recent deep-learning-based methods for protein function prediction. They presented an overview of current automated protein function prediction methods and conducted a mini comparison of several available tools. And they finally highlighted the challenges of the field. I have two other major concerns that are listed below:
1 Except for the deep-learning methods reviewed in this paper, several more advanced and recently published methods should be reviewed. Such as one in https://academic.oup.com/bioinformatics/advance-article/doi/10.1093/bioinformatics/btab198/6182677 that applied transformer and one in https://arxiv.org/abs/2007.12804 that applied GNN.
2 The data used as benchmark is not clearly explained in the Data section. I still don’t understand what does the NK (no-knowledge) and LK (limited-knowledge) mean. Does it mean the sequence has no or limited public annotations but was fully labeled first in the benchmark dataset? What do the partial mode and fully mode mean? The authors should explicit these two terms. I cannot understand the explanation from the current version from this sentence: “partial mode, for a set of proteins with at least one prediction, and full mode, computed for all benchmark proteins.”

Reviewer 2 ·

Basic reporting

1) Professional English used throughout
2) Background context and literature reference have been provided

Experimental design

Investigation into methods have been performed and results presented.
However, more structured details are required in the explanation of methods. While reading, details seemed confusing.

Validity of the findings

Conclusion have been supported by the results and overall review.

Additional comments

The review is well written giving an overview of conventional and new methods for protein function prediction.
Recommendation: Structured details are required in the explanation of methods.

---

## Round 0.2 · Minor Revisions

As you can see, the reviewer still thinks that the manuscript has some linguistic issues and requires additional editorial work. Please address these remaining concerns and make sure that the manuscript is edited by professional editors or fluent English speakers.

Reviewer 2 ·

Basic reporting

Background context and literature reference have been provided.

Experimental design

Including more details and flowchart presented helps in better understanding

Validity of the findings

The conclusion have been supported by the results

Additional comments

The manuscript is more structured and issues have been addressed. However few sections in the manuscript lacks correct sentence formation and is sometimes confusing. Working on improving those is recommended.

---

## Round 0.3 · accepted · Accept

As you can see, the reviewer was satisfied by your responses to the critiques and by revisions. Therefore, I am pleased to let you know that your amended manuscript is acceptable now.

Reviewer 2 ·

Basic reporting

Background provided, literature referenced, clear writing

Experimental design

Recommendations were worked on and methods described in details

Validity of the findings

The conclusion have been supported by the results

Additional comments

Recommendations have been worked on.